# Potential Application of Alternate Tillage (AT) in a Rice–Wheat Rotation System—Based on Soil Physical Properties, Wheat Growth and Yield

Shengchun Li, Yilin Zhang, Lihao Guo and Xiaofang Li *

College of Agriculture, Yangtze University, Jingzhou 434025, China
* Correspondence: lixiaofang20220607@163.com

**Abstract:** Alternate tillage (AT) has the potential to reduce inputs and improve soil quality and crop yield, but there has been no research on the effect of AT on soil and wheat in a rice–wheat rotation system. In this study, field experiments were conducted to examine the effects of four tillage management methods (conventional tilling (CT) in each crop (RCT–WCT), no tilling (NT) in rice and conventional tilling in wheat (RNT–WCT, AT1), conventional tilling in rice and no tilling in wheat (RCT–WNT, AT2), and no tilling in each crop (RNT–WNT)) on the physical properties of soil, wheat growth, and yield. At the 0–5 cm soil layer, CT in the wheat season increased bulk density (BD) and decreased total properties, but it decreased BD at the 5–40 cm soil layer, and the effect of RCT–WCT was significantly greater than that of RNT–WCT. CT in the wheat season increased the root activity, root dry weight, net photosynthetic rate, leaf area index, antioxidant enzyme activities, and yield, and there was no significant effect between RCT–WCT and RNT–WCT. RNT-WCT has the potential to reduce inputs and maintain wheat yields.

**Keywords:** no-tillage; alternate tillage; root function; photosynthetic capacity; oxidative balance





## 1. Introduction

Tillage is an important method of agricultural management that directly changes the physical properties of soil through agricultural machinery and indirectly affects soil nutrient availability, microbial and enzymatic activities, and crop growth [1,2]. Conventional tillage (CT) has made outstanding contributions to crop yield improvement through mediating air permeability, water infiltration, seedbed preparation, and weed control [1,3]. However, frequent CT decreases soil organic matter (SOM), the stability of aggregates, and soil fertility; increases soil erosion; and requires more labor [4,5]. No-tillage (NT) is considered a sustainable management practice that promotes soil health and system sustainability [6]. NT can reduce soil disturbance, improve soil structure, and provide better physical protection for SOC in soil aggregation, thus maintaining soil fertility [7,8]. However, studies have reported drawbacks of NT, such as stratification of soil nutrients, mechanical compaction, pH change, and increased weed pressure, all of which risk yield reduction [9,10]. Therefore, more environmentally friendly and productive sustainable tillage models need to be developed.

Wang et al. reported that strategic tillage, in which CT and NT are alternated at annual intervals, may be the sustainable tillage method favored in future, because it retains the advantages of both CT and NT [5]. The occasional use of CT in long-term NT farming systems has been repeatedly shown to maintain NT benefits while mitigating problems such as soil compaction, soil nutrient stratification, and weeds [11–13]. In addition, many scholars have studied the impact of strategic tillage practices on soils and crops by alternate tillage (AT) practices during each crop-growing season. Dai et al. [14] reported that, compared with CT, both AT (wheat-season tillage and corn-season no-tillage) and NT enhanced soil aggregation and increased the fungal population and organic carbon

content in soil macroaggregates, but the effect of NT was significantly greater than that of AT. Hu et al. [15] reported that, compared with CT, NT rather than AT (wheat-season tillage and corn-season no-tillage) significantly increased the external mycelium length, soil alkaline phosphatase activity, and soil organic C content. Pandey et al. [16] found that, compared with both tillage methods in a rice and wheat rotation system, the total and recalcitrant carbon contents and phenolics increased in both no-tillage in two-season and one-season no-till + another-season tillage. However, the effects of AT on soil and crops are influenced by factors such as soil properties, crop rotation patterns, and climatic conditions [16]. In the rice–wheat (RW) rotation system, one of the most important cropping systems in China, at least one rice crop is grown followed by one wheat crop on the same field each year [8]. Wheat is one of the most important crops in the world, and it is also the third most important crop in China in terms of planting area and yield [17].

Therefore, it is important and widely representative to study the effects of CT, NT, and AT on soil and wheat under the RW rotation system. With the above considerations in mind, this study compared the effects of CT, NT, and AT on soil properties, wheat root function, photosynthetic capacity, antioxidant capacity, and yield in an RW rotation system.

## 2. Materials and Methods

### 2.1. Experimental Site

Experiments were conducted in 2019 and 2020 in Libu Town (112°04′ N, 30°32′ E), Jingzhou City, Hubei Province, China. The area belongs to the northern subtropical agricultural climate zone, with annual average temperature of 16.5 °C, an accumulated temperature $\geq$10 °C of 5094.9–5204.3 °C, annual average precipitation of 1095 mm, and annual average sunshine time of 1718 h. We selected five points in the field, took soil samples of the 0–20 cm soil layer, and mixed the soil into one sample to determine the soil properties. The paddy soil was classified as a Grey Alluvial soil according to Chinese soil taxonomy. The soil texture is clay loam with 23% sand (2.00–0.02 mm), 41% silt (0.02–0.002 mm), and 36% clay (<0.002 mm), and the index of soil agrochemical properties was 5.83 pH, 29.44 g kg$^{-1}$ of organic matter, 214.52 mg kg$^{-1}$ of available N, 11.09 mg kg$^{-1}$ of available P, and 112.03 mg kg$^{-1}$ of available K. The cropping system was a rotation of summer rice (*Oryza sativa* L., HHZ) and winter wheat (*Triticum aestivum* L., ZM9023).

### 2.2. Experimental Treatments

A single-factor experimental design was adopted, with four tillage management methods: (T1) conventional tilling before transplantation/sowing of each crop (RCT–WCT), (T2) no tilling before transplantation of rice and conventional tilling before sowing of wheat (RNT–WCT), (T3) conventional tilling before transplantation of rice and no tilling before sowing of wheat (RCT–WNT), and (T4) no tilling before sowing of each crop (RNT–WNT). The experimental plot (5 m × 10 m) was arranged in random block arrangement, and each treatment was repeated three times. Specifically, RCT meant ploughing with a depth of 25 cm once at 10 days before transplanting, while the field was saturated with water by adjusting artificial irrigation 5 days before ploughing according to rainfall. Rotating was performed with a depth of 15 cm once at 5 days before transplanting and the base fertilizer was spread before the rotation, and then a water depth of 1–2 cm was maintained until transplanting. RNT meant that the soil was not disturbed after wheat harvest to before rice planting. The residue of the wheat covered the ground until rice harvest. The herbicide Gramoxone (paraquat 20%) was used (diluted 5 mL L$^{-1}$ and applied at 750 L ha$^{-1}$) 10 days before rice transplanting. At 5 days before planting, the water was maintained at a depth of 1–2 cm until transplanting and the basal fertilizer was spread on the soil surface 2 days before transplanting. WCT meant that the rotary tiller was deeply rotated twice with a depth of 20 cm, the base fertilizer was spread before the second rotation 2 days before sowing, and then the 2BJK-6 large-width precision seeder (Yuncheng Gongli Co., Ltd., Yuncheng, China) was used for sowing. WNT meant that the 2BMF-6 type no-tillage belt rotary seeder (Zhongjiang Zefeng Small Agricultural Machinery Manufacturing Co., Ltd.,

Zhongjiang, China) was used to complete the seeding and fertilization operation at one time. When sowing, only shallow rotary ditching was performed on the seeding belt, and the ditch width was 4–5 cm. The depth of the ditch was 3–5 cm, and the wheat seeds were sown into the ditch, and the soil and straw that have been ploughed could rebound and cover the seeds. Weed management was conducted using chlorpromazine emulsifiable oil, which contains 10% fenorim. Diseases, pests, and water were managed centrally to avoid yield losses.

N fertilizers of urea were applied to rice plants at a rate of 180 kg N ha$^{-1}$ (50%, 25%, and 25% at the seedling, tillering, and earring stages, respectively) and N fertilizers of urea were added to wheat plants at a rate of 144 kg N ha$^{-1}$ (50%, 30%, and 20% at the seedling, tillering, and boosting stages, respectively). Phosphate fertilizers of single superphosphate were applied to rice and wheat plants at 90 and 72 kg $P_2O_5$ ha$^{-1}$, respectively. Potassium fertilizers of potassium chloride were applied to rice and wheat plants at 180 and 144 kg $K_2O$ ha$^{-1}$, respectively. Crop stubble was left in the field for all treatments. Rice was sown on May 1, manually transplanted on June 1, and harvested in mid-October. The row spacing was 16 cm × 30 cm with two seedlings per hill. Wheat was seeded at a rate of 150 kg ha$^{-1}$ in mid-November and harvested in the following May.

*2.3. Measurement Items and Methods*

2.3.1. Wheat Yield and Yield Components

Wheat grain yields and panicle density were measured at maturity by taking 5 m$^2$ plant samples at the center of each plot. The filled grains in each 5 m$^2$ plant sample were separated from the straws. The filled grains were oven-dried at 70 °C to a stable weight and weighed, and grain yield was calculated at 14% moisture content. Plant samples (five hills) adjacent to the harvest area were taken for detection of yield components (spikelets per panicle and 1000-grain weight).

2.3.2. Bulk Density and Total Porosity of Soil

Soil samples were collected for all three replicates 7 days before the wheat harvest from 2019 to 2020. Soils were sampled to a depth of 40 cm, with samples taken from the 0–5 cm layer, 5–10 cm layer, 10–20 cm layer, and 20–40 cm layer. A standard cutting ring of 100 cm$^3$ was used to obtain soil samples which were then dried at 105 °C to determine bulk density (BD). The total porosity of the soil (TP) was evaluated using BD and mean particle density (2.65 g cm$^{-3}$) values [5].

2.3.3. Wheat Root Function

Wheat samples from each plot of the five hills were selected to measure the root dry weight and root activity at the mid-tillering stage and flowering stage. For each root sample, a cube of soil (25 cm in length × 16 cm in width × 20 cm in depth) around each individual hill was removed using a sampling core. Such a cube contains about 95% of total root biomass [18]. Plants from each plot of the five hills formed a sample at each measurement. The roots in each cube of soil were carefully rinsed with a hydropneumatic elutriation device (Gillison's Variety Fabrications, Benzonia, MI, USA). Portions of each root sample were used for measurement of root activity, while the other root samples were oven-dried at 70 °C to stable weights and then weighed. Root activity was determined by measuring oxidation of alpha-naphthylamine (α-NA) [18]. One gram of fresh roots was transferred into a 150 mL flask containing 50 mL of 20 ppm α-NA. The flasks were incubated for 2 h at room temperature in an end-over-end shaker. After that, the aliquots were filtered and 2 mL of aliquot was mixed with 1 mL of 1.18 mmol$^{-1}$ NaNO and 1 mL of sulfanilic acid and the resulting color was measured by a spectrophotometer.

2.3.4. Wheat Net Photosynthetic Rate and Leaf Area Index

Five wheat plants were selected in each plot to measure the net photosynthetic rate (Pn) and leaf area index (LAI) at the mid-tillering stage and flowering stage. The Pn

of the top fully expanded leaves on the main stem were determined by gas exchange analyzer (Li-6400, Li-COR Inc., Lincoln, NE, USA) between 9:30 a.m. and 11:00 a.m. when the photosynthetic active radiation above the canopy was 1200 mmol m$^{-2}$ s$^{-1}$. LAI was obtained as the area of fully expanded leaves on the plants divided by the rice planting area [19].

2.3.5. Determination of Antioxidant Activities and Malondialdehyde Content in Wheat Leaves

At the mid-tillering stage and flowering stage, fresh flag leaves from five plants for determination of Pn and LAI were collected and stored at $-80$ °C until biochemical analysis. The fresh flag leaf samples were grinded with liquid nitrogen to form homogenate. Then, 9 mL of 50 mM sodium phosphate buffer (pH 7.8) was added to the homogenate, and centrifuged at 8000× $g$ rpm for 15 min at 4 °C. The supernatant was gathered for determination of superoxide dismutase (SOD), peroxidase (POD), catalase (CAT) enzyme activities, and malondialdehyde (MDA) content, referring to Huang et al. [20]. One unit of SOD activity was the amount of enzyme that induced 50% inhibition in the initial rate of reduction of nitroblue tetrazolium at 560 nm. The mixture for determining POD consists of 1 mL of sodium phosphate buffer (pH 7.8), 0.95 mL of 0.2% guaiacol, 1 mL of 0.3% $H_2O_2$, and 0.05 mL aliquot of enzyme extract. The absorbance was read at 470 nm for 90 s with an interval of 30 s. One unit of POD activity was defined as the amount of enzyme that caused the decomposition of 1 mg substrate at 470 nm. The mixture for determining CAT consists of 1 mL of distilled water, 1 mL of 0.3% $H_2O_2$, and 0.05 mL enzyme extract. The absorbance was read at 470 nm for 90 s with an interval of 30 s. One unit of CAT activity was the decomposition of 1 M $H_2O_2$ at A240 within 1 min in 1 g of fresh leaves samples. The 1.5 mL enzyme extract was used to determine the content of MDA. The enzyme extract was mixed with 0.5 mL thiobarbituric acid solution prepared in 5% trichloroacetic acid and boiled at 100 °C for 30 min. The samples after cooling were centrifuged at 3000× $g$ rpm for 15 min. The absorbance was read at 450 nm, 532 nm, and 600 nm. The MDA contents were computed with the formula: MDA content = 6.45(OD$_{532}$ − OD$_{600}$) − 0.599OD$_{450}$.

*2.4. Statistical Analyses*

All experimental data were collected in 2019 and 2020 and expressed as mean ± standard error (SE) of three replicates. The normal distribution and homogeneity variance of data were tested using the Shapiro–Wilk test and Levene's test on SPSS 21.0 (IBM SPSS Statistics), respectively. Differences of indicators between 2 years (2019 and 2020) were compared by independent sample t-test. Differences in indicators between the four tillage methods were compared by one-way analysis of variance. In statistical analysis, two significance levels were set at $p < 0.05$ and $p < 0.01$. The histograms were drawn by Origin.

**3. Results**

*3.1. Effects of Tillage on Physical Properties of Soil*

Tillage has a significant effect on BD at 0–40 cm soil layer and TP at 0–5 cm soil layer, but has no significant effect on TP at 5–40 cm soil layer (Figure 1). Compared with RNT-WNT, BD at 0–5 cm soil layer increased by 2.18% in RCT-WNT, 10.50% in RNT-WCT, and 15.91% in RCT-WCT; BD at 5–10 cm soil layer decreased by 1.38% in RCT-WNT, 9.95% in RNT-WCT, and 12.06% in RCT-WCT; BD at 10–20 cm soil layer decreased by 5.76% in RCT-WNT, 8.22% in RNT-WCT, and 10.31% in RCT-WCT; and BD at 20–40 cm soil layer decreased by 4.80% in RCT-WNT, 10.58% in RNT-WCT, and 11.68% in RCT-WCT. Compared with RNT-RNT, TP at 0–5 cm soil layer decreased by 4.45% in RCT-WNT, 8.97% in RNT-WCT, and 10.31% in RCT-WCT.

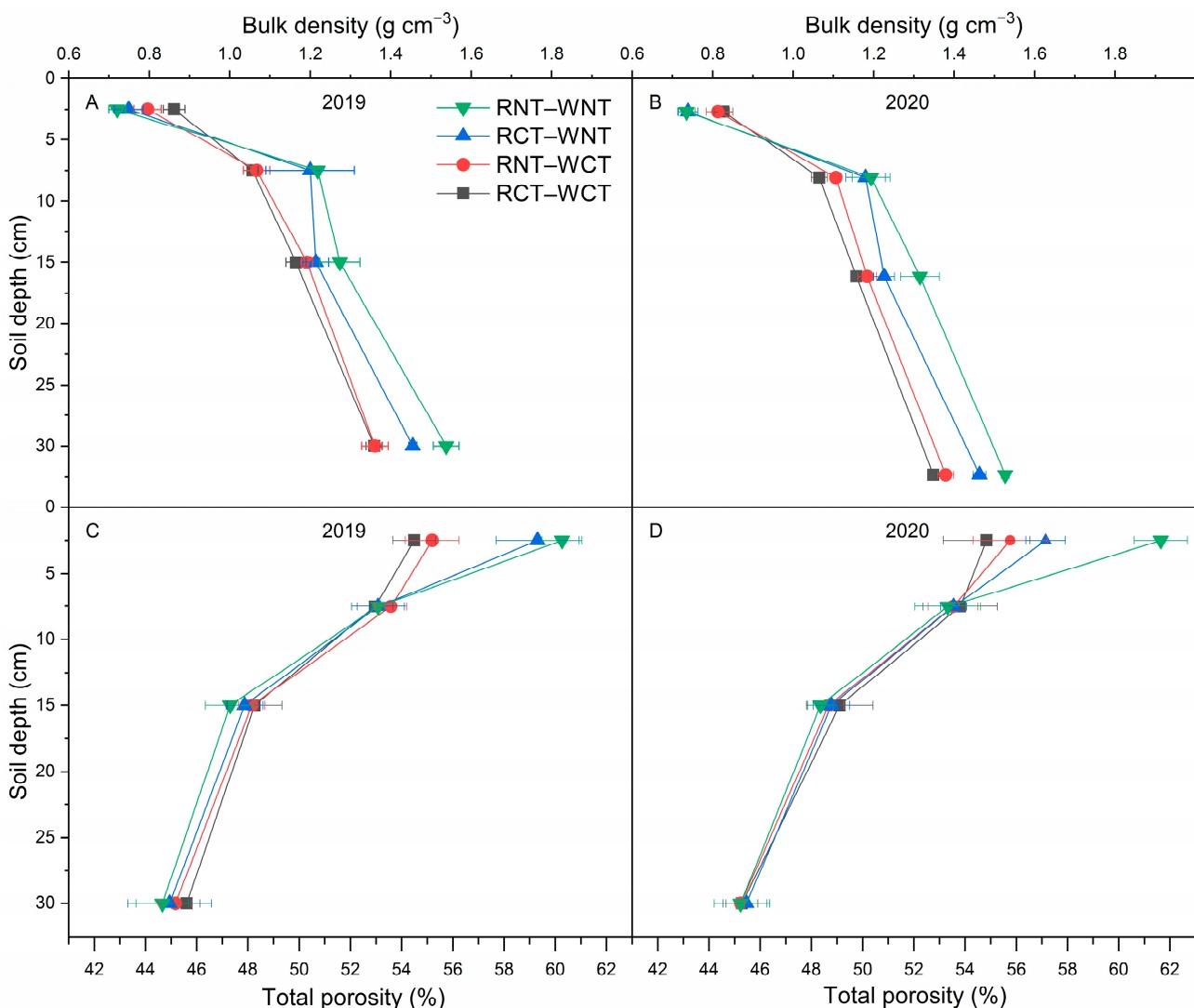

**Figure 1.** Effects of bulk density (BD, g m$^{-3}$) in 2019 (**A**) and 2020 (**B**) and total porosity (TP, %) in 2019 (**C**) and 2020 (**D**) under different tillage managements. RCT–WCT, conventional tillage before transplantation/sowing of each crop; RNT–WCT, no tillage before transplantation of rice and conventional tillage before sowing of wheat; RCT–WNT, conventional tillage before transplantation of rice and no tillage before sowing of wheat; RNT–WNT, no tillage before sowing of each crop. Different lowercase letters indicate statistical differences among treatments at *p* < 0.05.

### 3.2. Effects of Tillage on Root Activity and Root Dry Weight

Tillage has a significant effect on root activity and root dry weight both at the tiller stage and the flowering stage (Figure 2). Compared with RNT-RNT, root dry weight at the tiller stage increased by 2.18% in RCT-WNT, 27.18% in RNT-WCT, and 27.44% in RCT-WCT; root dry weight at the flowering stage increased by 4.66% in RCT-WNT, 34.06% in RNT-WCT, and 33.79% in RCT-WCT. Compared with RNT-RNT, root activity at the tiller stage increased by 3.73% in RCT-WNT, 32.40% in RNT-WCT, and 39.74% in RCT-WCT; root activity at the flowering stage increased by 4.18% in RCT-WNT, 33.79% in RNT-WCT, and 36.09% in RCT-WCT.

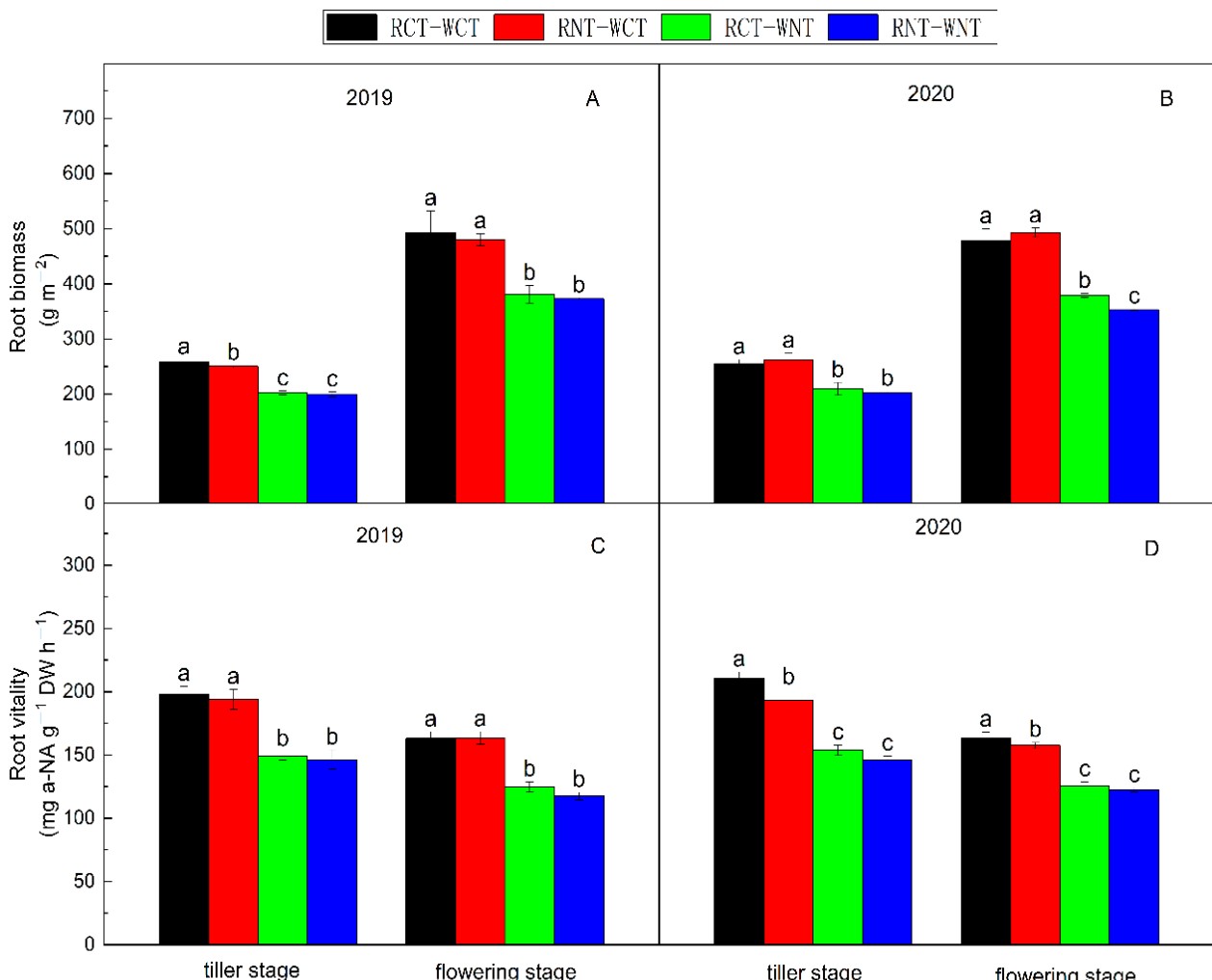

**Figure 2.** Root biomass in 2019 (**A**) and 2020 (**B**) and root vitality in 2019 (**C**) and 2020 (**D**) of rice under different tillage management methods. RCT–WCT, conventional tillage before transplantation/sowing of each crop; RNT–WCT, no tillage before transplantation of rice and conventional tillage before sowing of wheat; RCT–WNT, conventional tillage before transplantation of rice and no tillage before sowing of wheat; RNT–WNT, no tillage before sowing of each crop. Different lowercase letters indicate statistical differences among treatments at *p* < 0.05.

### 3.3. Effects of Tillage on Net Photosynthetic Rate and Leaf Area Index

Tillage has a significant effect on the net photosynthetic rate and leaf area index both at the tiller stage and the flowering stage (Figure 3). Compared with RNT-RNT, the net photosynthetic rate at the tiller stage increased by 0.69% in RCT-WNT, 7.30% in RNT-WCT, and 6.05% in RCT-WCT, whereas the net photosynthetic rate at the flowering stage increased by 3.40% in RCT-WNT, 6.42% in RNT-WCT, and 9.06% in RCT-WCT. Compared with RNT-RNT, leaf area index at the tiller stage increased by 4.00% in RCT-WNT, 15.18% in RNT-WCT, and 13.58% in RCT-WCT, whereas leaf area index at the flowering stage increased by 2.69% in RCT-WNT, 9.30% in RNT-WCT, and 14.15% in RCT-WCT.

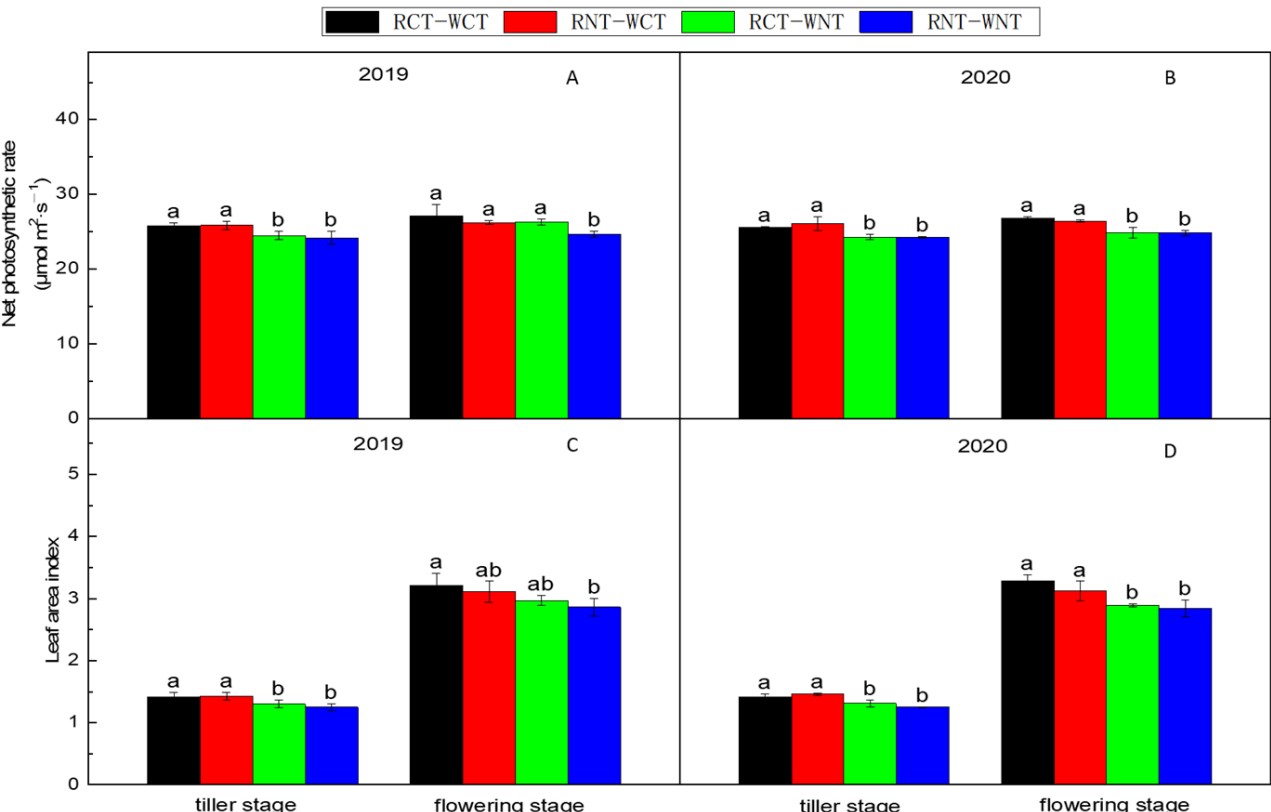

**Figure 3.** Net photosynthetic rate in 2019 (**A**) and 2020 (**B**) and leaf area index in 2019 (**C**) and 2020 (**D**) of rice under different tillage managements. RCT–WCT, conventional tillage before transplantation/sowing of each crop; RNT–WCT, no tillage before transplantation of rice and conventional tillage before sowing of wheat; RCT–WNT, conventional tillage before transplantation of rice and no tillage before sowing of wheat; RNT–WNT, no tillage before sowing of each crop. Different lowercase letters indicate statistical differences among treatments at $p < 0.05$.

### 3.4. Effects of Tillage on MDA Content and Activities of SOD, CAT, and POD

The year has a significant effect on MDA, SOD, CAT, and POD, except for CAT at the flowering stage (Figure 4). MDA content in 2019 was higher than that in 2020, and SOD, CAT, and POD activities were lower than those in 2020. Tillage has a significant effect on MDA, SOD, CAT, and POD at the flowering stage. Compared with RNT-RNT, MDA content decreased by 1.77% in RCT-WNT, 20.24% in RNT-WCT, and 19.65% in RCT-WCT; SOD activity increased by 4.98% in RCT-WNT, 30.18% in RNT-WCT, and 33.08% in RCT-WCT; CAT activity increased by 3.69% in RCT-WNT, 24.75% in RNT-WCT, and 32.64% in RCT-WCT; and POD activity increased by 2.01% in RCT-WNT, 23.60% in RNT-WCT, and 28.09% in RCT-WCT.

### 3.5. Effects of Tillage on Yield and Yield Composition

Tillage has a significant effect on grains per spike, 1000-grain weight, and yield (Table 1). Compared with RNT-RNT, grains per spike increased by 3.40% in RCT-WNT, 16.21% in RNT-WCT, and 18.20% in RCT-WCT; 1000-grain weight increased by 2.97% in RCT-WNT, 5.48% in RNT-WCT, and 8.07% in RCT-WCT; and yield increased by 6.18% in RCT-WNT, 27.83% in RNT-WCT, and 33.09% in RCT-WCT.

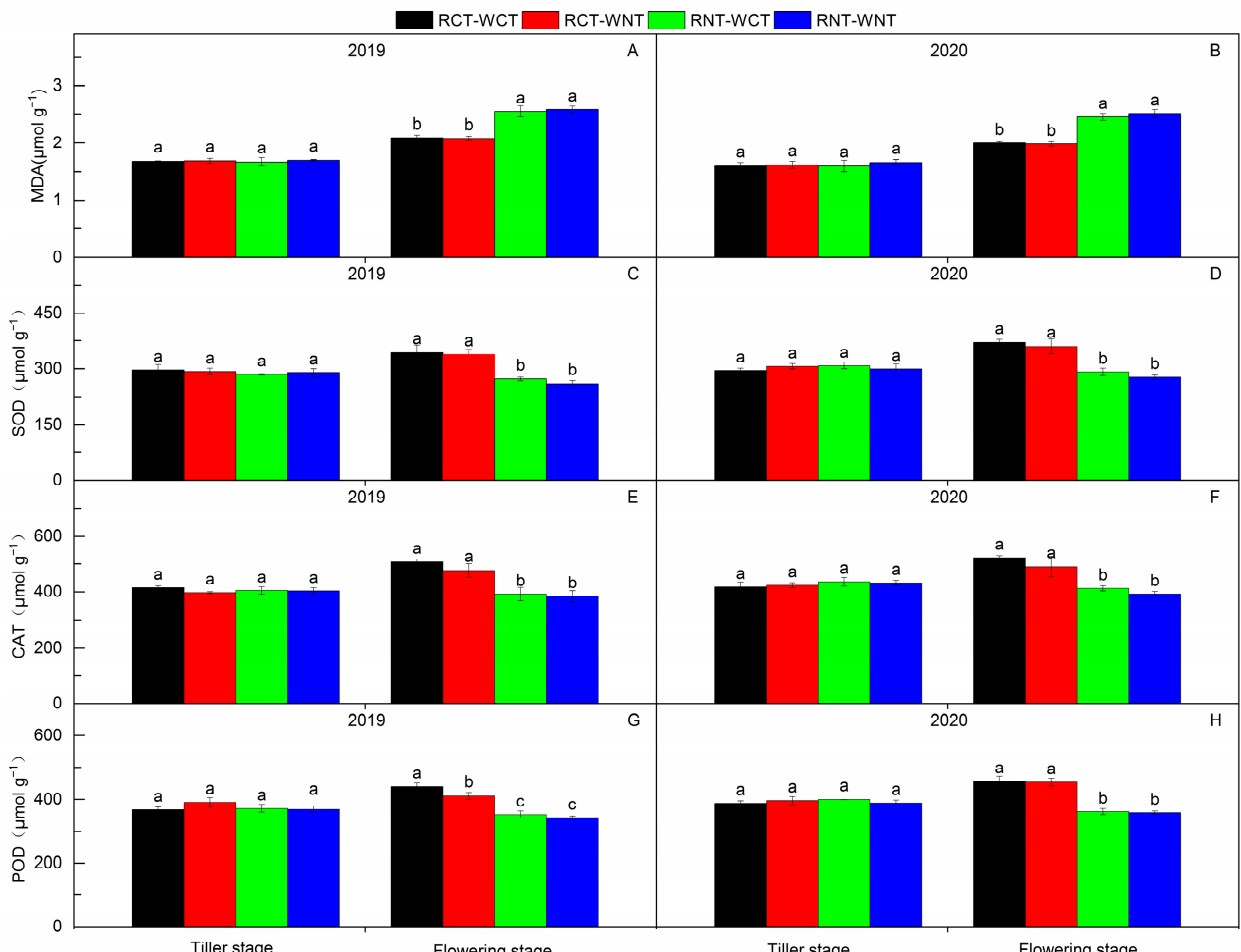

**Figure 4.** MDA in 2019 (**A**) and 2020 (**B**), superoxide dismutase (SOD) in 2019 (**C**) and 2020 (**D**), catalase (CAT) in 2019 (**E**) and 2020 (**F**), and peroxidase (POD) in 2019 (**G**) and 2020 (**H**) of rice under different tillage management methods. RCT–WCT, conventional tillage before transplantation/sowing of each crop; RNT–WCT, no tillage before transplantation of rice and conventional tillage before sowing of wheat; RCT–WNT, conventional tillage before transplantation of rice and no tillage before sowing of wheat; RNT–WNT, no tillage before sowing of each crop. Different lowercase letters indicate statistical differences among treatments at $p < 0.05$.

**Table 1.** Effects of the spike number $m^{-2}$, grains per spike, 1000-grain weight (g), and yield (t $ha^{-1}$) under different tillage treatments. RCT-WCT: conventional tilling before transplantation/sowing of each crop; RNT-WCT: no tilling before transplantation of rice and conventional tilling before sowing of wheat; RCT-WNT: conventional tilling before transplantation of rice and no tilling before sowing of wheat; RNT-WNT: no tilling before sowing of each crop. ns: no significant effects; *: significant effect at the $p < 0.05$ level; **: significant effect at the $p < 0.01$ level. Those values within a column among the same year followed by the same letter are not significantly different according to Duncan's test (0.05).

| Year | Tillage | Spike Number $m^{-2}$ | Grains per Spike | 1000-Grain Weight | Yield |
|---|---|---|---|---|---|
| | RCT-WCT | 466.04 ± 16.45a | 33.97 ± 0.99a | 47.12 ± 0.61a | 7.29 ± 0.56a |
| | RNT-WCT | 466.02 ± 21.68a | 33.23 ± 1.93a | 46.86 ± 0.43a | 7.1 ± 0.72a |
| 2019 | RCT-WNT | 451.51 ± 31.3a | 29.18 ± 0.83b | 44.85 ± 1.17b | 5.81 ± 0.53b |
| | RNT-RNT | 439.48 ± 17.29a | 28.38 ± 0.66b | 43.39 ± 0.27c | 5.31 ± 0.34b |

**Table 1.** *Cont.*

| Year | Tillage | Spike Number m$^{-2}$ | Grains per Spike | 1000-Grain Weight | Yield |
|---|---|---|---|---|---|
| | RCT-WCT | 463.37 ± 13.22a | 33.12 ± 0.34a | 47.59 ± 0.35a | 7.15 ± 0.26a |
| 2020 | RNT-WCT | 464.17 ± 41.5a | 32.73 ± 1.12a | 45.58 ± 0.62b | 6.77 ± 0.8a |
| | RCT-WNT | 436.73 ± 15.09a | 29.51 ± 0.79b | 45.39 ± 0.82b | 5.71 ± 0.15b |
| | RNT-RNT | 451.29 ± 11.97a | 28.38 ± 0.63b | 44.25 ± 1.01b | 5.54 ± 0.25b |
| Year | | * | ** | * | ** |
| Tillage | | ** | ** | ** | ** |
| Year * Tillage | | ns | ns | ns | ns |

## 4. Discussion

### 4.1. Bulk Density and Total Porosity of Soil

Tillage can improve the structure and ventilation conditions of topsoil, and this potential effect varied in the different tillage systems [21]. In this study, at the 0–5 cm soil layer, NT in wheat season decreased BD and increased TP (Figure 1), which is similar to results reported in other studies [22,23]. NT covers the topsoil with straw and increases the organic matter content of the topsoil, resulting in low bulk density and higher porosity [2]. In addition, as a result of soil biological activity under NT, increased aggregation and permanent pore development resulted in an increase in the number of macro-aggregates and the total and effective porosity. This, in turn, led to greater infiltration and crop water availability [22]. Similar to other studies [5,24], the soil BD in the 5–40 cm soil layer was increased under NT in wheat season, which shows that although NT reduces the BD of the topsoil, it increases the BD of the deep soil layer. It was also found in our unpublished rice season data that NT in the rice season resulted in a decrease in BD in the 0–5 cm soil layer and an increase in BD in the 5–40 cm soil layer in rice season soil. There was no difference in TP at the 10–40 cm soil layer. However, previous reports suggested that tillage affects TP [25]. The reason may be that tillage only affected the TP of the topsoil or tillage has a time-sensitive effect on TP.

### 4.2. Root Function and Photosynthetic Capacity

Root function (root dry weight and root activity) and photosynthetic capacity (Pn and LAI) are important indicators of rice growth [26]. Generally, the root function and photosynthetic ability at the mid-tillering stage were in accordance with RCT-WCT ≈ RNT-WCT > RCT-WNT ≈ RNT-WNT (Figure 2). Contrary to our results, Yang et al. [27] reported that NT increased root proliferation and N acquisition, activated the release of plant-available soil N and P, and improved LAI and plant dry mass production, conserved soil moisture, and improved wheat tillering and yield in a dry climate. Guo et al. [28] reported that NT provided a source of photosynthesis by increasing leaf area index and leaf area duration across the entire growth period of wheat to increase grain yield. Notably, their study was conducted under a corn–wheat rotation system. Our experiments were in a rice–wheat rotation system in which alternating wet and dry irrigation management during the rice growing season resulted in soil compaction, and NT exacerbated the negative impact of compaction on wheat root growth and photosynthesis. Under the rice–wheat rotation system, wheat season tillage can break the compacted soil, increase soil permeability (Figure 1), promote root development and nutrient and water absorption, and improve photosynthetic capacity. Similarly, Guan et al. [29] reported that tillage practice could reduce soil bulk density and penetration resistance and promoted root development and yield formation. In addition, the corn–wheat rotation system is usually applied in areas with less rainfall, and NT can reduce water evaporation and promote wheat root growth and photosynthesis. However, our test site had sufficient rainfall and had a low dependence on NT to maintain field moisture.

### 4.3. MDA Content and Antioxidant Responses

MDA is an important indicator of the degree of plant stress, and SOD, POD, and CAT are important antioxidant enzymes in plants [30]. NT in the wheat season increased the MDA content and decreased the activities of antioxidant enzymes in leaves during the flowering period, indicating that the NT in the wheat season caused stress to the growth of the wheat during the flowering period. Compact soils limit crop root development (Figures 1 and 2), hindering root uptake of nutrients and water [31]. However, NT did not affect the redox system of wheat leaves in the mid-tillering stage. The reason may be that the root system of wheat in the mid-tillering stage has just begun to develop downward [18] and is less stressed by soil obstruction [32].

### 4.4. Yield and Yield Composition

In previous studies, the effect of NT on wheat yield was controversial. Zhang et al. [33] reported that grain yields of wheat and maize within a given residue management practice were not significantly higher for reduced/no-tillage than continuous tillage, regardless of the effects of tillage on aggregates and soil nutrients. Zhang et al. [34] showed that no tillage had no significant effect on rice and wheat yield after 7 years in East China. Omara et al. [35] reported that NT increased the soil organic carbon, total soil nitrogen, and wheat yield. Shao et al. [36] reported that NT helped to improve available P and available K in topsoil (0–20 cm) and increased water use efficiency and wheat yield. Ding et al. [37] reported that NT decreased in N leaching, increased soil available N and root N uptake, and improved wheat grain yield and N utilization efficiency. However, many studies were carried out in arid regions, and the increase of NT in terms of wheat yield was attributed to NT reducing water loss and improving soil water storage [38–41]. Effects of tillage on wheat yield varied by species, region, and agronomic and environmental factors [42]. In this study, wheat yield was ranked as RCT-WNT $\approx$ RCT-WCT > RNT-WCT $\approx$ RNT-WNT (Table 1). Our data show that tillage increases wheat yield by reducing soil compaction, promoting root and photosynthesis, increasing antioxidant capacity, and promoting wheat growth in a rice–wheat rotation system. In addition, CT improved spike number m$^{-2}$, grains per spike, and 1000-grain weight, indicating that compared with NT, CT promoted yield formation in all critical periods of wheat. It was also found in our unpublished rice season data that tillage in rice season promoted rice growth and yield formation. There is no significant difference in yield between RCT-WCT and RNT-WCT, indicating that it is feasible to reduce one tillage under the premise of maintaining wheat yield, but for the sustainable development of a rice–wheat rotation system, more rice yield data are needed.

## 5. Conclusions

RNT-WCT has an excellent effect on improving soil quality and promoting wheat growth and yield based on the benefit of saving one tillage. It is a more economical and efficient tillage mode, but it needs more rice season data to support it.

**Author Contributions:** S.L. and Y.Z. designed the experiments, S.L. and L.G. analyzed the data and wrote the article, S.L. and Y.Z. performed the traits investigation, and S.L. and X.L. revised and edited the manuscript. S.L. and Y.Z. have contributed equally to this work. All authors have read and agreed to the published version of the manuscript.

**Funding:** This research was funded by the National Natural Science Foundation of China (31071482) and Ministry of Finance (Agriculture) Industry Special (201303008).

**Institutional Review Board Statement:** Not applicable.

**Informed Consent Statement:** Not applicable.

**Data Availability Statement:** The data presented in this study are available on request from the corresponding author. The data are not publicly available because the relevant test results have not been published.

**Conflicts of Interest:** The authors declare no conflict of interest.

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
