# Peer review of "Potential Application of Alternate Tillage (AT) in a Rice–Wheat Rotation System—Based on Soil Physical Properties, Wheat Growth and Yield"

_soilsystems, doi:10.3390/soilsystems6030070_

Round 1
Reviewer 1 Report
I appreciate this work, as it deals with an interesting issue of explaining strageties of alternative tillage and its effects on the soil and the crops, with particular concern with the annual rice-wheat rotation. The work archived the local practices of tillage and management of the soil and crops, as well as the respected outputs and impacts on the cropping system. Yet some minor revisions are needed.
1. How the rice and the wheat is rotated locally?
2. Is the 20 days or more flooding period the natural process or management induced stage? What is its reason or consideration? Is this flooding fallow period prefered for the rice-wheat rotation?
3. How the RNT is performed? with what machine and how the standing stubble and the wheat residue is treated? with or without interference with rice seedling transplanting?
4. It's difficult to understand the relationship between "tilling to 25 cm" and "sampling to 40 cm".
5. Typewriting errors were found in a number of places.
Author Response
Thank you very much for reviewing our manuscript. Your comments are very important to improve the quality of the article and our future study. Under the guidance of your comments, we have completed the first revision of the manuscript. Thank you again and wish you good health and success in your career.
- How the rice and the wheat is rotated locally?
Reply:Rice was sown on May 1, transplanted on June 1 and harvested in mid-October. The row spacing was 16 cm × 30 cm with two seedlings per hill. Wheat was seeded at a rate of 150 kg ha−1 in mid-November and harvested in the following May.
- Is the 20 days or more flooding period the natural process or management induced stage? What is its reason or consideration? Is this flooding fallow period prefered for the rice-wheat rotation?
Reply:Irrigation of 4-5cm of water 20 days before rice transplanting is a one-time use, the purpose is to moisten the soil before deep plowing, and 20 days is an approximate time. We have revised this sentence to make the meaning more accurate
- How the RNT is performed? with what machine and how the standing stubble and the wheat residue is treated? with or without interference with rice seedling transplanting?
Reply:We added RNT to handle more detailed operations. Including weed control methods and rice transplanting methods. Rice is transplanted manually.
- It's difficult to understand the relationship between "tilling to 25 cm" and "sampling to 40 cm".
Reply:The reserved depth of ploughing is 25cm. Since the machine has an error of about 5cm, we measured the relevant indicators of the 20-40cm soil layer.
- Typewriting errors were found in a number of places.
Reply:We applied for retouching services
Reviewer 2 Report
1. Language, especially tenses requite attention.
2. Introduction is brief and concise.
3. Materials and methods require improvement:
Line 65: Only two seasons. Personally I am not comfortable with only two seasons where soil tillage is used and especially from depths of 5 - 40 cm. Upper layer (0-5 cm) is acceptable. When the effect of tillage alone is evaluated then the results are in general obvious. If the authors attempted to compare the systems within the two consecutive seasons, then I accept the outcome.
Line 69: Provide the physical properties (texture) e.g. percentage sand, clay and silt.
Line70-72: Provide extraction methods.
Under Experimental treatments the layout of the experiment was not provided. Number of replications (could be deduced from line 117) as well as plot size are required.
Line 105: "row spacing was 16x30 cm²" does not make sense. Rephrase.
Line 139: "Five planting pits of wheat" not clear! Does this mean five plants were selected?
Discussion is generally brief but clear (concise).
General comments and suggestions:
Tenses should be corrected by an English linguist.
The authors do not comply with the correct use of SI units e.g. 25cm instead of 20 cm and 15 °C instead of 20°C.
Line 360 surnames of authors in uppercase instead of lower case.
With reference to figure I suggest that the legend be placed below the graph above the caption.
Author Response
Thank you very much for reviewing our manuscript. Your comments are very important to improve the quality of the article and our future study. Under the guidance of your comments, we have completed the first revision of the manuscript. Thank you again and wish you good health and success in your career.
- Language, especially tenses requite attention.
Reply:We applied for retouching services
- Introduction is brief and concise.
- Materials and methods require improvement:
Line 65: Only two seasons. Personally I am not comfortable with only two seasons where soil tillage is used and especially from depths of 5 - 40 cm. Upper layer (0-5 cm) is acceptable. When the effect of tillage alone is evaluated then the results are in general obvious. If the authors attempted to compare the systems within the two consecutive seasons, then I accept the outcome.
Reply: Dear reviewers, we did continuously measure plant and soil data for 4 seasons (2 wheat seasons + 2 rice seasons), but the data for rice seasons were measured by another graduate student in our group. Therefore, rice data is published by him and meets his postgraduate graduation requirements. Since his article on the rice season has not yet been published, we cannot provide you with information on the rice season in our article. We can only add to the discussion " It was also found in our unpublished rice season data that NT in rice season resulted in a decrease in BD in 0–5 cm soil layer and an increase in BD in 5–40 cm soil layer in rice season soil.” and “It was also found in our unpublished rice season data that tillage in rice season promoted rice growth and yield formation”
Line 69: Provide the physical properties (texture) e.g. percentage sand, clay and silt.
Reply: We supplemented the physical properties of the soil, including the percentage of sand, clay and silt.
Line70-72: Provide extraction methods.
Reply: We have supplemented the collection process of soil samples
Under Experimental treatments the layout of the experiment was not provided. Number of replications (could be deduced from line 117) as well as plot size are required.
Reply:We supplemented the arrangement of the experimental plots, dealing with the number of repetitions and plot size
Line 105: "row spacing was 16x30 cm²" does not make sense. Rephrase.
Reply:We have corrected this typo
Line 139: "Five planting pits of wheat" not clear! Does this mean five plants were selected?
Reply:We changed "Five planting pits of wheat plants" to "Five wheat plants" meaning five plants were selected
Discussion is generally brief but clear (concise).
General comments and suggestions:
Tenses should be corrected by an English linguist.
Reply:Done
The authors do not comply with the correct use of SI units e.g. 25cm instead of 20 cm and 15 °C instead of 20°C.
Reply:We have corrected these formats
Line 360 surnames of authors in uppercase instead of lower case.
Reply:We have corrected this format
With reference to figure I suggest that the legend be placed below the graph above the caption.
Reply:We have readjusted the layout of graphs and tables
Reviewer 3 Report
The article is of an important theme, but some improves are needed. More attention to the historical data of the area and climate. More comments in the attached file.

Author Response
Thank you very much for reviewing our manuscript. Your comments are very important to improve the quality of the article and our future study. Under the guidance of your comments, we have completed the first revision of the manuscript. Thank you again and wish you good health and success in your career.
Line 26,30. We explain what CT and NT mean and how they operate in the experimental design.
Line 64. We supplemented the meteorological data during the experiment.
Line 69:We have added more historical information on the area, including soil properties and the percentage of sand, clay and silt.
Line 72:In China, there is more rainfall in summer. In China's rice-wheat rotation, rice is usually grown in summer.
Line 74:We supplemented the number of replicates for the trial treatment
Line 179:We have corrected this typo
Line 188:We have supplemented the legend to Figure 1
Line 268:We supplemented the meteorological data during the experiment.
Line 323:We have rewritten the conclusion to make it more concise.
